# Simultaneous Determination of Enantiomeric Purity and Organic Impurities of Dexketoprofen Using Reversed-Phase Liquid Chromatography—Enhancing Enantioselectivity through Hysteretic Behavior and Temperature-Dependent Enantiomer Elution Order Reversal on Polysaccharide Chiral Stationary Phases

**DOI:** 10.3390/ijms25052697

**Published:** 2024-02-26

**Authors:** Máté Dobó, Gergely Dombi, István Köteles, Béla Fiser, Csenge Kis, Zoltán-István Szabó, Gergő Tóth

**Affiliations:** 1Department of Pharmaceutical Chemistry, Semmelweis University, Hogyes 9, H-1092 Budapest, Hungary; dobo.mate@stud.semmelweis.hu (M.D.); dombi.gergely@stud.semmelweis.hu (G.D.); koteles.istvan@pharma.semmelweis-univ.hu (I.K.); 2Department of Chemistry and Molecular Biology, University of Gothenburg, Medicinaregatan 19, 41390 Göteborg, Sweden; 3Institute of Chemistry, University of Miskolc, H-3515 Miskolc, Hungary; 4Ferenc Rakoczi II. Transcarpathian Hungarian College of Higher Education, 90200 Beregszasz, Ukraine; 5Department of Physical Chemistry, Faculty of Chemistry, University of Lodz, 90-149 Lodz, Poland; 6Department of Pharmaceutical Industry and Management, George Emil Palade University of Medicine, Pharmacy, Science and Technology of Targu Mures, Gh. Marinescu 38, 540139 Targu Mures, Romania; csenge.kis@gmail.com (C.K.); zoltan.szabo@umfst.ro (Z.-I.S.); 7Sz-imfidum Ltd., Lunga nr. 504, 525401 Targu Mures, Romania

**Keywords:** chiral separation, chiral switch, HPLC, ketoprofen, method optimization, polysaccharide chiral column, retention curve, thermodynamic

## Abstract

A reversed-phase high-performance liquid chromatographic (HPLC) method was developed for the simultaneous determination of the potential impurities of dexketoprofen, including the distomer R-ketoprofen. After screening the separation capability of four polysaccharide columns (Lux Amylose-1, Lux Amylose-2, Lux Cellulose-1 and Lux Cellulose-2) in polar organic and in reversed-phase modes, appropriate enantioseparation was observed only on the Lux Amylose-2 column in an acidified acetonitrile/water mixture. A detailed investigation of the mobile phase composition and temperature for enantio- and chemoselectivity showed many unexpected observations. It was observed that both the resolution and the enantiomer elution order can be fine-tuned by varying the temperature and mobile phase composition. Moreover, hysteresis of the retention times and enantioselectivity was also observed in reversed-phase mode using methanol/water mixtures on amylose-type columns. This could indicate that the three-dimensional structure of the amylose column can change by transitioning from a polar organic to a reversed-phase mode, which affects the enantioseparation process. Temperature-dependent enantiomer elution order and rare enthalpic/entropic controlled enantioseparation in the operative temperature range were also observed in reversed-phase mode. To find the best methodological conditions for the determination of dexketoprofen impurities, a full factorial optimization design was performed. Using the optimized parameters (Lux Amylose-2 column with water/acetonitrile/acetic acid 50/50/0.1 (*v*/*v*/*v*) at a 1 mL/min flow rate at 20 °C), baseline separations were achieved between all compounds within 15 min. Our newly developed HPLC method was validated according to the current guidelines, and its application was tested on commercially available pharmaceutical formulations. According to the authors’ knowledge, this is the first study to report hysteretic behavior on polysaccharide columns in reversed-phase mode.

## 1. Introduction

A chiral switch strategy involves the redevelopment of a chiral drug that has already been approved as a racemate, with emphasis on developing and marketing a single enantiomer. Chiral switches offer a strategic advantage by harnessing the unique properties of individual enantiomers. This approach ensures that only the most pharmacologically active form of a drug reaches the market, optimizing therapeutic efficacy and patient safety [1,2,3]. One of the classic examples of a chiral switch is omeprazole. An important class for chiral switch is proton pump inhibitors, where the marketed individual enantiomers present a more favorable pharmacokinetic profile than the racemate, leading to improved clinical outcomes for patients with hyperacidity-related diseases and extended patent protection for pharmaceutical companies [4]. The other examples are non-steroidal anti-inflammatory drugs (NSAID), like ibuprofen and ketoprofen [5]. Due to its 2–4 times greater potency compared to the racemate, dexketoprofen was incorporated into therapy as a single-enantiomeric drug [6,7]. Analytical innovations are essential for chiral switching, as the transformation of the non-effective form into an impurity post-switching requires its detection and quantification at an extremely low concentration. In addition, all other chemical impurities must also be determined. Pharmacopeial monographs and the pharmaceutical industry traditionally utilize separate analytical techniques to quantify chemical and enantiomeric impurities. However, adopting methods that can simultaneously measure both chiral and chemical impurities can result in significant time and cost savings. An effective approach to streamline the analysis of enantiomeric purity and related substances is by employing a single chiral column in HPLC [8,9,10,11].

In the field of chiral analysis, the use of a chiral stationary phase (CSP) in direct HPLC methods is widely recognized as the preferred approach. Among the multitude of chiral selectors available on the market, polysaccharide-type chiral selectors are the most employed. Their key advantage lies in their multimodal nature, allowing them to be used in normal-phase, reversed-phase, hydrophilic interaction liquid chromatography (HILIC) and polar organic (PO) modes [12,13]. Furthermore, these columns also find extensive application in chiral supercritical fluid chromatography (SFC) [14]. One notable drawback of the normal-phase mode is the use of potentially harmful solvents, such as hexane. In contrast, the PO mode employs neat alcohols and acetonitrile (ACN) or mixtures of these solvents, significantly reducing the environmental impact. This mode is particularly well suited for preparative purposes [8]. Undoubtedly, the reversed-phase mode, utilizing water-based mobile phases, could be considered the most environmentally friendly system. However, it is interesting to note that reversed-phase enantioseparation is relatively underrepresented in the literature, especially in pharmacopeias [15,16]. Also, some of the recently described phenomena have not been investigated in the reversed-phase mode yet. For example, it is known that different alcohols (2-propanol vs. methanol (MeOH)) can alter the three-dimensional structure of some of the amylose-based selectors, which results in different enantiorecognition capacity under PO and normal-phased modes. Horváth and Simon showed that seemingly identical chromatographic conditions can result in dramatically different enantioseparations on columns containing amylose tris(3,5-dimethylphenylcarbamate) chiral selector in mixtures of 2-propanol and methanol. Selectivity towards a structurally diverse set of enantiomer pairs dependeds on the direction from which the composition of the eluent is approached. The observed phenomenon, known as hysteresis, was later demonstrated to be a common occurrence on amylose-type stationary phases, observable in numerous polar organic eluent mixtures [17,18,19]. To date, hysteresis has been documented only in normal-phase and PO modes, with no exploration in reversed-phase mode. It is noteworthy that the presence of hysteresis across different PO mixtures can prove advantageous during method development. In eluent mixtures, the amylose-based chiral selector may exist in multiple conformational states, each offering distinct enantiorecognition mechanisms with potential success under different circumstances [17,20]. This discovery holds the potential to usher in a novel, more accessible and cost-effective approach to chiral method development, shifting the focus from stationary phase-centered methods to those centered around the mobile phase composition [21].

Observing the enantiomer elution order (EEO) is the starting point to gather information on the interaction between the analyte and different CSPs. EEO reversals suggest different enantiorecognition mechanisms under different conditions. In recent years, there have been numerous studies regarding EEO reversals on polysaccharide-type CSPs. These occurred upon changes in the structure of the chiral selector, mobile phase constituents, the type and concentration of acidic or basic additives, as well as column temperature [22,23,24,25]. It is unequivocal that different chiral selectors can lead to differences in enantiorecognition. As mentioned earlier, the mobile phase can affect the three-dimensional structure of the chiral selector, which, in turn, can modify the EEO. Besides this, solvents can alter the interactions between the CSP and analyte. The type and concentration of additive can modify the protonation state of the analyte, which is crucial for the enantiorecognition mechanism. The effect of temperature is more complicated. Temperature influences both thermodynamic and kinetic aspects of the enantiomer complexation process occurring on the CSP. The effects of temperature on retention and enantioseparation factors are unpredictable and are in particular need of evaluation [26,27,28].

Numerous papers in the existing literature focus on the chiral separation of ketoprofen enantiomers using HPLC and diverse chiral selectors, such as cinchona alkaloid- [29], polysaccharide- [30,31,32] or protein-based [33] CSPs. However, within these articles, only a recent study by Tok et al. focuses on evaluating the enantiomeric purity of dexketoprofen [31]. Notably, there is currently no literature available that simultaneously addresses the separation of the chiral and chemical impurities of dexketoprofen.

The objective of our recent article is to pioneer an innovative, environmentally friendly approach to the simultaneous determination of R-ketoprofen, ketoprofen impurity C (3-(1-carboxyethyl)benzoic acid), ketoprofen impurity A (1-(3-Benzoylphenyl)ethenone) and ketoprofen ethyl ester (ethyl 2-(3-benzoylphenyl)propanoate) (Figure 1) in a S-ketoprofen sample with an acceptance criterion of no more than 0.1% for each impurity. Additionally, we delve into the influence of mobile phase composition and temperature on the enantioseparation of dexketoprofen in reversed-phase mode, employing a polysaccharide CSP. A thermodynamic analysis and the temperature-dependent reversal of EEO is investigated in detail. The hysteresis of retention and enantioselectivity on polysaccharide-based columns in reversed-phase mode is explored for the first time in the literature.

## 2. Results and Discussion

### 2.1. Scouting Phase

As an initial step in method development, the chiral separation of ketoprofen enantiomers was undertaken, representing a crucial aspect of the separation challenge at hand. Two amylose-based chiral columns, Lux Amylose-1 and Lux Amylose-2, along with two cellulose-based columns, Lux Cellulose-1 and Lux Cellulose-2, were evaluated in both PO and reversed-phase modes. The chiral selectors in Lux Amylose-1 and Lux Cellulose-1 have the same 3,5-dimethylphenylcarbamate substituent, while Lux Amylose-2 and Lux Cellulose-2 feature a chlorine-containing chiral selector (Appendix A). The results of the scouting phase screening are presented in Table 1.

Notably, the PO mode failed to achieve enantioseparation on the four CSPs examined, and cellulose-based CSPs exhibited no chiral recognition. In contrast, Lux Amylose-2 exhibited high chiral recognition (with R_s_ = 3.32) in an acidified ACN/water mixture, while Lux Amylose-1 showed a smaller resolution (with R_s_ = 0.67) in an acidified MeOH/water mixture. An intriguing aspect to consider is the influence of water content on retention. In a MeOH/water mixture, an increase in water content resulted in higher retention. However, in an ACN/water mixture, a distinct trend was observed: at low water concentrations, the retention time decreased, but from 30% water onward, it began to increase. Above 30% water content in the MeOH/water mixture, there was a notable increase in analysis time, and similarly in the ACN/water mixture above 50% water content. Consequently, we refrained from exploring additional mixtures in the scouting phase. Moreover, an appropriate result was also obtained with a very good resolution. It is evident that the quantity of water has a significant impact on enantioselectivity. On the Lux Amylose-2 column, increasing the water content from 30% to 50% resulted in the resolution increasing from 0 to 3.32. Furthermore, it was noted that cellulose-type columns in polar organic mode exhibited higher retention factors, although interestingly, chiral separation was not achieved.

It can be seen that baseline separation is exclusively exhibited by the Lux Amylose-2 column with a mobile phase composition of ACN/water/acetic acid of 50:50:0.1 (*v*/*v*/*v*). Consequently, additional method developments were conducted under these specific conditions.

### 2.2. Method Development

To find the optimal parameter ranges, first, a “one factor at a time” screening approach was applied with all the impurities covered, tracking the obtained critical resolution values within the following chromatographic parameter ranges: a column temperature of 10–50 °C, a flow rate between 0.5 and 1 mL/min, acetic acid content between 0% and 0.15% and mobile phase composition between 40 and 70% ACN content in water. Adding acetic acid (or formic acid) as an acidic additive to the mobile phase was necessary for enantioseparation; however, a concentration of higher than 0.1% did not have a significant influence on separation performance. The flow rate did not influence the resolution; however, a higher flow rate resulted in lower retention time and better peak shapes, but higher flow rate than 1 mL/min cannot be used due to the high increase in back pressure.

For further method optimization, a full factorial design was used, where the parameters which influence separation to a higher extent were optimized, namely mobile phase composition (between 40 and 70 ACN *v*/*v*% in water) and temperature (between 10 °C and 50 °C). Resolution values and analysis time were selected as response variables. The experimental matrix and the obtained results are summarized in Table 2. The table presents the original input data with two parallel sets, which were shuffled randomly to ensure robustness.

Our goals were to baseline-separate all compounds with a resolution of at least 1.5 for every peak to have an ideal EEO of R > S (distomer first EEO) and to have a reasonable analysis time (under 15 min). It should be noted that after the peak of dexketoprofen, there are still peaks, but in this case, the resolution values are generally above 5, so the critical resolutions are between enantiomers. The order of the investigated compounds is highly dependent on the applied circumstances. The parameters were set to maximize the resolutions and simultaneously minimize the analysis time. Maximizing the critical resolution, Rs_2_ had the highest weight of all, while all other dependent variables were set to be middle-weighted. A polynomial model was meticulously applied, and a comprehensive analysis of variance (ANOVA) was conducted to assess the significance of our model. When evaluating the performance indicators of the regression models, we considered the R^2^ and R^2^_adj_ values. The R^2^ values were 0.9943, 0.9834, 0.9477, 0.9851 and 0.9432, while the R^2^_adj_ values were 0.9926, 0.9782, 0.9315, 0.9805 and 0.9256 for analysis time at R_s1_, R_s2_, R_s3_ and R_s4_, respectively. In each instance, the consistency between the R^2^ and R^2^_adj_ values indicates the suitability of the models for effective navigation within the experimental domain. Both investigated parameters (temperature and ACN%) have a significant impact on both resolution and analysis time.

Appendix A illustrates three-dimensional response surface plots constructed based on the regression models. To determine the optimal combination of analytical conditions for both responses, the statistical software’s optimization feature was employed utilizing the Derringer desirability function.

The optimal analytical conditions, identified through the desirability function, were as follows: Lux Amylose-2 column, ACN/water/acetic acid 50:50:0.1 (*v*/*v*/*v*), flow rate 1 mL/min, column temperature 20 °C. Using these optimal analytical conditions, the baseline resolution of all compounds was achieved within 15 min (Figure 2A). We examined and analyzed the impact of temperature and mobile phase content on enantiomer separation in more detail, and we describe it in Section 2.4 and Section 2.5.

### 2.3. Method Validation and Application

The method was validated according to guideline Q2 (R1) of the International Council for Harmonization for all chemical impurities, and for R-ketoprofen as a chiral impurity with respect to sensitivity, linearity, accuracy and precision. The validation data are summarized in Table 3.

The method’s sensitivity was assessed by evaluating the limits of detection (LOD) and quantification (LOQ). These limits were determined using a signal-to-noise ratio of 3:1 for LOD and 10:1 for LOQ. The validation of impurity determination was conducted within the range of 0.05–0.3% regarding a target concentration of 5 mg/mL of dexketoprofen. The linearity of the method was established at six concentration levels within the specified range for all impurities. Calibration plots indicated a linear relationship, with correlation coefficients exceeding 0.9979 in all cases. Additionally, the 95% confidence intervals of the y-intercepts included zero, and the residuals showed random distributions.

Accuracy and precision were assessed through intraday and interday (intermediate precision) evaluations involving five replicate injections at three concentration levels for all impurities, encompassing the linear range. Injections were conducted on the same day and repeated on two consecutive days. All data can be found in Table 3.

The accuracy for all impurities, expressed as average recovery %, ranged from 98.18% to 100.33%. Intraday precision, represented by RSD%, fell within the range of 0.12% to 2.00%, while the RSD for intermediate precision was below 2.11%. The validation results affirm that the method is sensitive, linear, accurate and precise for determining the selected impurities in dexketoprofen.

The optimized and validated method was employed for analyzing two real pharmaceutical samples, specifically, film-coated tablets containing a nominal content of 25 mg dexketoprofen. Representative chromatograms for the two tablets are depicted in Figure 2B,C, respectively, while the impurity content of the investigated products is summarized in Table 4.

Both investigated products were found to contain distomer impurity and impurity A in low amounts, along with other unidentified impurities, while impurity C and ketoprofen ethyl ester were not detected. Notably, both products showed elevated levels of R-ketoprofen impurity, surpassing 0.1%. This discovery aligns with the results reported by Tok et al. [22]. The level of impurity A was observed to be below 0.2% (<0.05% based on our measurement), and the total amount of all impurities did not exceed 1%, in accordance with the pharmacopeia requirement outlined in the ketoprofen monographs. It is crucial to note that dexketoprofen lacks a monograph, and consequently, the regulatory authority has yet to establish the limit for R-ketoprofen in this context.

### 2.4. The Role of Eluent in Mixture in Chiral Recognition—Hysteresis Phenomenon in PO and Reversed-Phase Mode

It is widely recognized that the mobile phase plays a pivotal role in influencing enantioseparation among chiral selectors. Specifically, the choice of eluent significantly impacts the enantiorecognition capability of a CSP. In modern chromatography, the utilization of amylose-type CSPs introduces a noteworthy phenomenon known as retention and enantioselective hysteresis. This phenomenon offers advantages stemming from the chiral selector’s eluent-dependent stable conformational states [17,21,34,35]. Nevertheless, it is essential to acknowledge the potential drawbacks associated with hysteresis. The different conformational states induced by the eluent can lead to distorted results if the storage eluent differs from the eluent used during measurements or proper column equilibration is not achieved. Therefore, careful consideration of these factors is imperative to ensure the accuracy and reliability of chromatographic outcomes [17,19].

MeOH/water, ACN/water and ACN/MeOH mixtures were investigated during the enantioseparation of ketoprofen on four cellulose and amylose columns (Lux Amylose-1, Lux Amylose-2, Lux Cellulose-1, Lux Cellulose-2). Using our method, we systematically increased the quantity of one eluent component by 10 *v*/*v*%. At a certain point, we conducted a reverse-direction measurement, enabling us to analyze the retention and selectivity profile, including hysteresis. In the MeOH/water mixture, the retention profile differed from that in the ACN/water and ACN/MeOH mixtures, regardless of the column used. For the MeOH/water mixture, a classical reversed-phase retention pattern was observed: with an increase in water content, a continuous increase in retention time was re-corded (Figure 3A). In the other two cases, mixed-mode reversed-phased-HILIC elution behavior was noted, indicated by a U-shaped retention curve. However, the course of the retention curve varied, as the presence of water had a greater impact on retention in the water/ACN mixture compared to the effect of MeOH in the MeOH/ACN mixture (Figure 3B,C). This difference in elution curves could be attributed to the fact that, unlike MeOH, ACN is an aprotic solvent.

As expected from previous works [17,18,21,34], a considerable hysteresis effect was observed on the Lux Amylose-1 column, a minor hysteresis effect was observed in the Lux Amylose-2 column and no (or not relevant) hysteresis occurred on the cellulose-based columns. Hysteresis has already been described in PO mode, such as in the MeOH/ACN mixture [19,34]. In this study, hysteresis was also observed in reversed-phase mode, especially in the MeOH/water mixture for the first time (Figure 3D and Figure 4). Interestingly, the hysteresis effect in the ACN/water mixture was negligible. This can be explained by the fact that the higher-order structure of the amylose also exists in different, stable conformers in reversed-phase mode, which causes the hysteresis effect. However, the reversion of the conformation state in the MeOH/water mixture is slower, or the change is greater from the outset. Our study also shows that the same polysaccharide column can be used in reversed-phase mode and in PO mode, and it is not necessary to allocate a separate column for these methods.

### 2.5. Thermodynamic Study—Temperature-Dependent Elution Order Reversal

The temperature-dependent reversal of elution order is one of the rarest observed changes in enantiomeric elution order in the literature, even though, according to the classical van ’t Hoff theory, there always exists a temperature point where the two enantiomers would co-elute, and the EEO is different below and above this temperature point [36,37]. Temperature-dependent reversal of the EEO of ketoprofen was observed earlier in the work of Matarashvili et al. on a Lux amylose-1 column using a classic normal-phase eluent [38].

In our work to investigate the impact of temperature on retention and enantioselectivity, the column temperature was changed from 10 to 55 °C on a Lux Amylose-2 column under reversed-phase conditions using a water/ACN/acetic acid 50/50/0.1 (*v*/*v*/*v*) mixture. Interestingly, in the studied temperature range, enantiomeric elution order reversal was observed, as shown in Figure 5.

At 40 °C, the two enantiomers co-eluted; below 40 °C, the enantiomeric elution order was R > S; and above this temperature, S > R. For detailed thermodynamic analysis, van ’t Hoff analysis was applied [39,40]. The van ’t Hoff plots for retention and enantioseparation factors are depicted in Appendix A. The calculated thermodynamic parameters are summarized in Table 5.

The ln(α) (selectivity) vs. 1/T plot exhibits a characteristic V shape, revealing a minimum at 40 °C (calculated as 40.67 °C and 40.21 °C). This marks the isoenantioselective temperature, where the two enantiomers co-elute due to a balance between entropy and enthalpy effects. Below 40 °C, enantioseparation is enthalpy-controlled; above 40 °C, enantioseparation becomes entropy-driven, indicating that increased temperature results in enhanced enantioselectivity. The measured and calculated isoenantioselective temperatures are in good agreement.

## 3. Materials and Methods

### 3.1. Materials

Dexketoprofen ((2S)-2-(3-benzoylphenyl)propanoic acid; CAS: 22161-81-5), racemic ketoprofen (2-(3-benzoylphenyl)propionic acid; CAS: 22071-15-4), ketoprofen impurity C (3-(1-carboxyethyl)benzoic acid); CAS: 68432-95-1), ketoprofen impurity A (1-(3-Benzoylphenyl)ethenone); CAS: 66067-44-5), ketoprofen ethyl ester (ethyl 2-(3-benzoylphenyl)propanoate); CAS: 60658-04-0), ACN, formic acid, and acetic acid were obtained from Sigma-Aldrich, Hungary (Budapest, Hungary). Deionized water was prepared using a Millipore Milli-Q water purification system (Burlington, MA, USA). Chiral columns, namely Lux Amylose-1 [Amylose tris(3,5-dimethylphenylcarbamate)], Lux Amylose-2 [Amylose tris(5-chloro-2-methylphenylcarbamate)], Lux Cellulose-1 [Cellulose tris(3,5-dimethylphenylcarbamate)] and Lux Cellulose-2 [Cellulose tris(3-chloro-4-methylphenylcarbamate)] (150  ×  4 mm, particle size 5 μm) were obtained from Phenomenex (Torrance, CA, USA). Dekenor (KRKA, d. d. Slovenia) and Ketodex tablets (BERLIN-CHEMIE AG, Germany), both containing a stated amount of 25 mg of dexketoprofen, were purchased from a local pharmacy in Budapest, Hungary.

### 3.2. HPLC Analysis

An LC-UV analysis was carried out on two different systems: an Agilent 1100 HPLC system, consisting of an inline degasser (G1322A), a quaternary pump (G1311A), an automatic injector (G1329A) paired with sample thermostat (G1330A), a column thermostat (G1316A) and a diode array detector (G1315A) with Agilent Chemstation B04.03-SP2 software, and an Agilent 1260 Infinity HPLC system (G1312B binary gradient pump, G1367E autosampler, G1315C diode array detector) with Agilent MassHunter B.03.01 software (Agilent Technologies, Waldbronn, Germany).

In the scouting phase, the enantioseparation capability of the applied four polysaccharide-type HPLC columns (Lux Amylose-1, Lux Amylose-2, Lux Cellulose-1, Lux Cellulose-2) was screened in PO and reversed-phase mode. In PO mode, ACN and MeOH modified with 0.1% acetic acid were applied, respectively. In reversed-phase mode, the following eluent compositions were applied: MeOH/water/acetic acid 90/10/0.1 (*v*/*v*/*v*), MeOH/water/acetic acid 70/30/0.1 (*v*/*v*/*v*), ACN/water/acetic acid 90/10/0.1 (*v*/*v*/*v*), ACN/water/acetic acid 70/30/0.1 (*v*/*v*/*v*) and ACN/water/acetic acid 50/50/0.1 (*v*/*v*/*v*). The chromatographic conditions were kept constant during screening: flow rate was 1.0 mL/min, column temperature was kept at 20 °C, injection volume was 1 μL. Whenever an experiment required pretreatment, 10 column volumes of the corresponding eluent were flushed through the column.

The hysteresis study was performed in PO and reversed-phase modes, respectively, based on the guidelines of our previous works using all four columns (Lux Amylose-1, Lux Amylose-2, Lux Cellulose-1, Lux Cellulose-2) [13,14]. In PO mode, the MeOH/ACN mixtures were investigated, going from 100 *v*/*v*% MeOH to 100 *v*/*v*% ACN and back to 100 *v*/*v*% MeOH in 10 *v*/*v*% increments. Both organic eluents were mixed with 0.1 *v*/*v*% acetic acid. In reversed-phase mode the ACN/water and MeOH/water mixtures were investigated. Measurements went from 100% organic eluent to 30% organic eluent and back to 100% organic eluent in 10% increments. Every eluent contained 0.1 *v*/*v*% acetic acid. During the hysteresis study, chromatographic conditions were constant at 1 mL/min flow and a 20 °C column temperature. The ketoprofen sample was prepared as follows: a 1 mg/mL stock solution was prepared using MeOH as a solvent from the racemic ketoprofen and from dexketoprofen separately. A total of 100 μL from the racemic stock solution was spiked with 50 μL of the dexketoprofen solution, and then, diluted to a final volume of 1000 μL using MeOH, resulting in a concentration of 50 μg/mL of R(−)-ketoprofen and 100 μg/mL of S(+)-ketoprofen.

During the method development for the simultaneous determination of chiral and achiral impurities, stock solutions of 200 μg/mL dexketoprofen were prepared in MeOH and were spiked with all (chemical and enantiomeric) impurities at around 2% (around 4 μg/mL). Using Stat-Ease Design-Expert v7.0.0 software (Stat-Ease, Minneapolis, MN, USA), a full factorial design was set, where ACN *v*/*v*% in the eluent and column temperature were independent variables. The final test solution of dexketoprofen used for method validation and applicability measurement was 5 mg/mL. The impurity level percentages were calculated relative to this concentration (e.g., 0.1% impurity level means 5 μg impurity in 5 mg dexketoprofen sample. The detection wavelength was 230 nm.

### 3.3. Analysis of Pharmaceutical Formulation

Both tablets (Dekenor and Ketodex) were prepared the same way: ten tablets were weighed and pulverized in a mortar. The accurately weighed portion of tablet powder corresponding to about 50 mg dexketoprofen was extracted in 10.0 mL of MeOH with 0.1 *v*/*v*% acetic acid medium using an ultrasonic bath for 30 min. Afterwards, the samples were centrifuged for 5 min, applying 4000 rpm (MiniSpin, Eppendorf, Germany), and filtered with a 0.22 μm polytetrafluoroethylene (PTFE) membrane filter (Millipore Corp, Cork, Ireland). The resulting solutions contained a concentration of 5 mg/mL of dexketoprofen and were examined using the optimized HPLC method.

### 3.4. Thermodynamic Study

The effect of the column temperature on chromatographic separation was investigated between 10 and 55 °C on the Lux Amylose-2 CSP using ACN/water/acetic acid 50:50:0.1 (*v*/*v*/*v*) and a 1 mL/min flow rate. A classical van ’t Hoff analysis was carried out to gain a better understanding of the energetic interactions. van ’t Hoff plots were constructed by plotting the natural logarithm of the retention factor as a function of the inverse of the absolute temperature.
(1)$$k=\frac{\Delta H{}^{\circ}}{RT}+\frac{\Delta S}{R}+ln\Phi \;$$
where *R* stands for the universal gas constant; *T* is the temperature in Kelvin; *k* is the retention factor of the individual enantiomers; Δ*H*° denotes the standard enthalpy, while Δ*S*° is the standard entropy change in the transfer of the solute from the mobile phase to the stationary phase; and *Φ* is the phase ratio of the Lux Amylose-2 column. If Δ*H*° is constant in the selected temperature range, a linear relationship is obtained between *lnk* and 1/T, with a slope of −Δ*H*°/*R* and an intercept of Δ*S*°/*R* + *lnΦ*. Since the value of the phase ratio is seldom known, Δ*S*°* values (Δ*S*°* = Δ*S*° + *RlnΦ*) are often used to compensate for the uncertainty in *Φ*.

Similarly, the differences in the change in standard enthalpy Δ(Δ*H*°) and standard entropy Δ(Δ*S*°) for the two enantiomers moving from the mobile phase to the stationary phase were also calculated according to modified van ’t Hoff equation:(2)$$ln\alpha =-\frac{\Delta \left(\Delta H{}^{\circ}\right)}{RT}+\frac{\Delta (\Delta S{}^{\circ})}{R}$$

The isoenantioselective temperatures (*T_iso_*) were determined by calculating the ratio between Δ(Δ*H*°) and Δ(Δ*S*°) as follows:(3)$$T_{iso}=\frac{∆(∆H{}^{\circ})}{∆(∆S{}^{\circ})}$$

*T_iso_* represents the temperature at which the enthalpy contribution is precisely balanced by the entropic term, resulting in a Gibbs free energy (Δ(Δ*G*°)) value of zero.
(4)∆(∆G°)=∆(∆H°)−T∗∆(∆S°)

The thermodynamic parameter Δ(Δ*G*°) offers insights into the binding strength between the analyte and selector, with more negative values indicating a more effective binding interaction. Consequently, when Δ(Δ*G*°) equals zero, it signifies that there is no disparity in the binding strength between the enantiomers. Therefore, at *T_iso_*, the two enantiomers co-elute, and no separation is achieved. Above and below *T_iso_*, the enantiomeric elution order is different.

## 4. Conclusions

Our research focused on the development and optimization of an HPLC method for the simultaneous determination of potential impurities in dexketoprofen, with a particular emphasis on enantioseparation. Through a meticulous screening process of four polysaccharide columns, we identified Lux Amylose-2 as the most effective column for achieving the desired enantioseparation in an acidified ACN/water mixture. The investigation into mobile phase composition and temperature revealed unexpected and nuanced observations. Notably, the resolution and EEO were found to be finely tunable by varying the temperature and mobile phase composition. The hysteresis of retention and enantioselectivity, particularly in reversed-phase mode using the MeOH/water mixture, hinted at a a potential impact of the amylose CSP tridimensional structure transition between the polar organic and reversed-phase modes. Of particular interest was the discovery of temperature-dependent enantiomer elution order and rare enthalpic/entropic-controlled enantioseparation within the operative temperature range in reversed-phase mode. This phenomenon adds a layer of complexity to the understanding of chiral separations under different conditions. To determine the optimal method for dexketoprofen impurity determination, we employed a full factorial optimization design. Using the identified optimal parameters, including the Lux Amylose-2 column with 0.1% (*v*/*v*) acetic acid in an ACN/water 50/50 (*v*/*v*) mixture at a flow rate of 1 mL/min and a temperature of 20 °C, baseline separations were achieved within a short 15 min timeframe. The newly developed HPLC method underwent rigorous validation according to current guidelines and demonstrated its applicability to commercially available pharmaceutical formulations. Our study not only contributes to the refinement of chromatographic methodologies for pharmaceutical analysis, but also reveals intriguing and previously undiscovered aspects of enantioseparation behavior, particularly the observed hysteric behavior in reversed-phase mode. These findings pave the way for further exploration and understanding of the intricacies of chiral separations in HPLC.

## Figures and Tables

**Figure 1 ijms-25-02697-f001:**
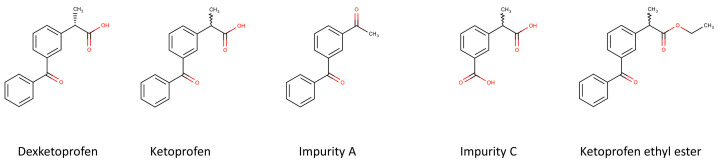
The structures of ketoprofen and ketoprofen impurities.

**Figure 2 ijms-25-02697-f002:**
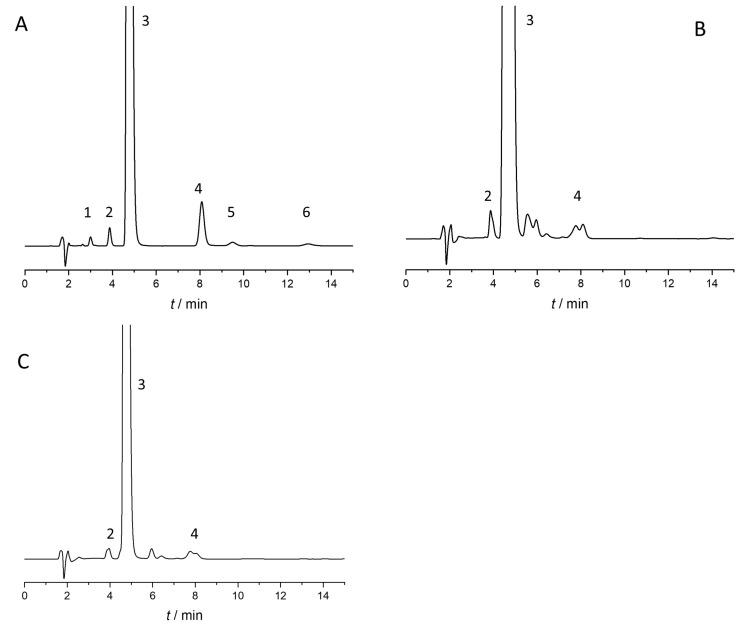
Representative chromatograms obtained during method optimization and application. (**A**) Dexketoprofen sample spiked with 0.1% enantiomeric impurity. (**B**) Solution of Dekenor 25 mg tablet. (**C**) Solution of Ketodex 25 mg tablet. Chromatographic conditions: Lux Amylose-2 column with water/ACN/acetic acid 50/50/0.1 (*v*/*v*/*v*), 1 mL/min flow rate at 20 °C (1—impurity C, 2—R-ketoprofen, 3—S-ketoprofen, 4—impurity A, 5 and 6—racemic ketoprofen ethyl ester).

**Figure 3 ijms-25-02697-f003:**
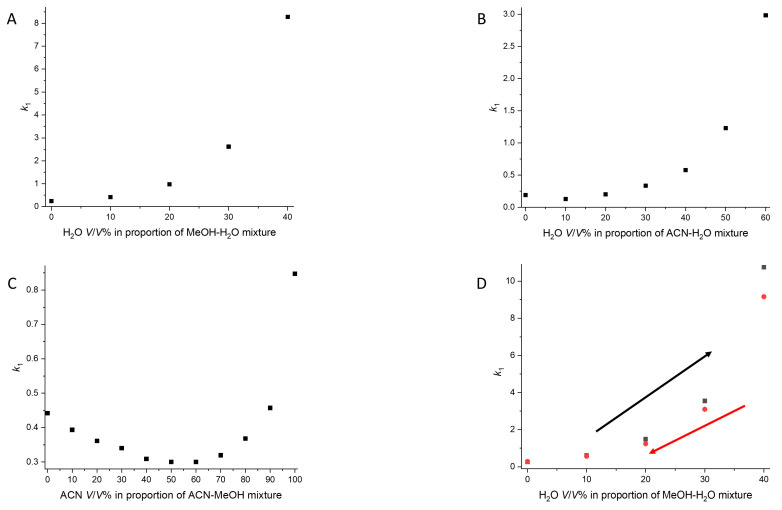
The effect of eluent on separation. (**A**) Plot of the retention factors of the first-eluting enantiomer (k**_1_**) as a function of the water content in MeOH. (**B**) Plot of the retention factors of the first-eluting enantiomer (k**_1_**) as a function of the water content in ACN. (**C**) Plot of the retention factors of the first-eluting enantiomer (k**_1_**) as a function of ACN content in MeOH. (**D**) Plot of the retention factors of the first-eluting enantiomer (k**_1_**) as a function of the water content in MeOH during the hysteresis study.

**Figure 4 ijms-25-02697-f004:**
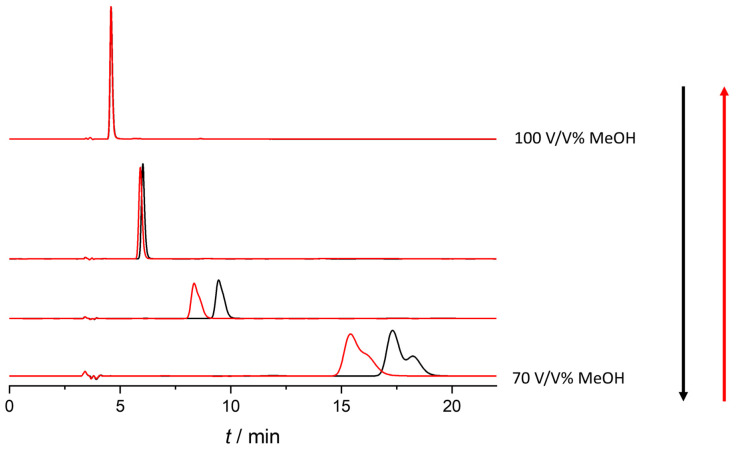
Enantioseparation of ketoprofen in different MeOH/water mixtures during the hysteresis study on Lux amylose-1 column (flow rate: 1 mL/min; 20 C; the mobile phases contain 0.1% acetic acid).

**Figure 5 ijms-25-02697-f005:**
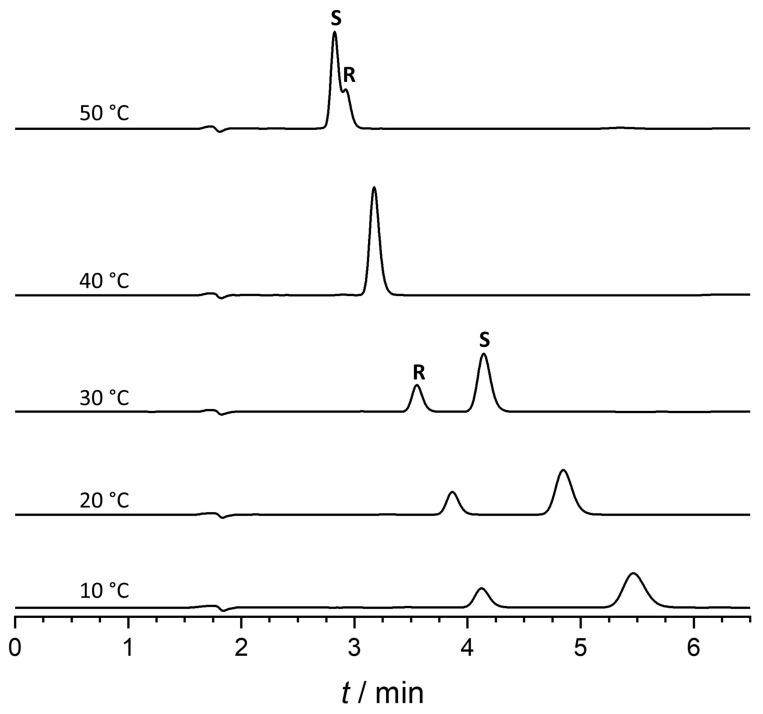
Temperature-dependent EEO reversal of ketoprofen on Lux amylose-2 column (mobile phase: water/ACN/acetic acid 50/50/0.1 (*v*/*v*/*v*), flow rate: 1 mL/min).

**Table 1 ijms-25-02697-t001:** Chromatographic data obtained during the preliminary study related to the retention factor of the first-eluting enantiomer (k_1_) and resolution (Rs).

Column	Mobile Phase *	k_1_	R_s_
Lux Amylose-1	MeOH	0.26	0
	ACN	0.19	0
	MeOH/water 90:10	0.61	0
	MeOH/water 70:30	3.55	0.67
	ACN/water 90:10	0.13	0
	ACN/water 70:30	0.33	0
	ACN/water 50:50	1.23	0
Lux Amylose-2	MeOH	0.29	0
	ACN	0.75	0
	MeOH/water 90:10	0.53	0
	MeOH/water 70:30	2.49	0
	ACN/water 90:10	0.51	0
	ACN/water 70:30	0.93	0
	ACN/water 50:50	1.77	3.32
Lux Cellulose-1	MeOH	0.63	0
	ACN	0.93	0
	MeOH/water 90:10	0.94	0
	MeOH/water 70:30	4.01	0
	ACN/water 90:10	0.33	0
	ACN/water 70:30	0.62	0
	ACN/water 50:50	1.98	0
Lux Cellulose-2	MeOH	0.74	0
	ACN	1.43	0
	MeOH/water 90:10	1.03	0
	MeOH/water 70:30	3.92	0
	ACN/water 90:10	0.63	0
	ACN/water 70:30	1.13	0
	ACN/water 50:50	3.57	0

* with 0.1% acetic acid.

**Table 2 ijms-25-02697-t002:** Full factorial design table used for method optimization with the obtained results for the selected responses.

Run No.	ACN in Water (%) * (Factor 1)	Temperature (°C) (Factor 2)	Analysis Time (min)	R_s1_ **	R_s2_ **	R_s3_ **	R_s4_ **	Elution Order ***
1	70	20	4.1	1.44	0	4.78	0.87	C > R = S > A > ester
2	60	30	5.5	0	2.02	7.27	1.79	C = R > S > A > ester
3	50	10	13.9	4.55	3.46	7.56	4.42	C > R > S > A > ester
4	50	20	12.5	5.07	3.32	8.91	4.64	C > R > S > A > ester
5	70	40	3.5	1.16	0	4.85	0.8	C > R = S > A > ester
6	60	20	6.3	1.89	2.42	6.63	2.41	C > R > S > A > ester
7	60	10	7.0	1.88	2.72	06.01	2.30	C > R > S > A > ester
8	70	10	4.4	1.94	0	4.39	0.91	C > R = S > A > ester
9	50	30	10.3	3.45	2.48	9.97	3.91	C > R > S > A > ester
10	50	40	7.9	3.30	0	11.30	1.64	C > R = S > A > ester
11	60	40	5.1	1.36	0	7.31	2.55	C > R = S > A > ester
12	70	30	3.6	1.3	0	4.58	0	R > C = S > A = ester
13	70	30	3.7	1.33	0.8	4.71	0	R > C = S > A = ester
14	70	40	3.5	1.14	0	4.87	0.79	C > R = S > A > ester
15	60	30	5.5	0	2,17	7.27	1.8	C = R > S > A > ester
16	60	20	6.3	1.88	2.4	6.67	2.38	C > R > S > A > ester
17	60	10	7.0	1.91	2.7	5.98	2.30	C > R > S > A > ester
18	70	10	4.4	1.97	0	4.39	0.91	C > R = S > A > ester
19	50	20	12.4	5.04	3.32	8.86	4.59	C > R > S > A > ester
20	50	40	7.8	3.04	0	11.35	1.47	C > R = S > A > ester
21	50	10	13.9	4.62	3.46	7.59	4.42	C > R > S > A > ester
22	50	30	10.4	3.64	2.5	10.01	3.97	C > R > S > A > ester
23	60	40	5.1	1.13	0	7.50	2.55	C > R = S > A > ester
24	70	20	4.1	1.48	0	4.74	0.87	C > R = S > A > ester
25	40	10	45.2	11.56	4.43	8.59	08.07	C > R > S > A > ester
26	40	10	45.3	11.23	4.38	8.64	8.01	C > R > S > A > ester
27	40	20	37.6	11.63	4.33	10.43	8.53	C > R > S > A > ester
28	40	20	37.9	11.99	4.42	10.42	08.02	C > R > S > A > ester
29	40	30	29.6	10.09	3.44	12.56	7.44	C > R > S > A > ester
30	40	30	29.5	9.95	3.46	12.49	7.29	C > R > S > A > ester
31	40	40	18.8	7.23	0	16.24	03.08	C > R = S > A > ester
32	40	40	18.3	6.65	0	16.44	2.72	C > R = S > A > ester
33	40	50	14.4	5.10	0	16.55	2.12	C > R = S > A > ester
34	40	50	14.4	4.93	0	16.47	02.06	C > R = S > A > ester
35	70	50	2.7	0	1.78	1.55	1.08	R = S > C > ester > A
36	70	50	2.7	0	1.62	1.85	1.21	R = S > C > ester > A
37	60	50	3.7	0	0.85	6.96	1.62	R = S > C > ester > A
38	60	50	3.6	0	0.79	6.96	1.62	R = S > C > ester > A
39	50	50	6.2	0.85	0	10.72	03.09	C > R = S > A > ester
40	50	50	6.2	0.90	0	10.72	03.09	C > R = S > A > ester

* with 0.1% acetic acid. ** R_s1_, R_s2_, R_s3_ and R_s4_ represent the resolution between the first and second peaks, between the second and third peaks, between the third and fourth peaks and between the fourth and fifth peaks, respectively. *** C = Ketoprofen impurty C, R = R-ketoprofen, S = S-ketoprofen, A = ketoprofen impurity A, ester = ketoprofen-ester impurity.

**Table 3 ijms-25-02697-t003:** Summary of data obtained during method validation for the simultaneous determination of related substances and enantiomeric purity in a 5 mg/mL dexketoprofen sample.

Parameter	Level	IMP-A	R-ket	IMP-C	Ester-a	Ester-b
Range (%)		0.05–0.3	0.05–0.3	0.05–0.3	0.1–0.3	0.1–0.3
Equation		743.6x + 6.965	654.51x + 1.896	1738.2x + 7.822	560.43x + 2.108	612x + 1.123
r^2^		0.9988	0.9990	0.9979	0.9991	0.9994
LOD (μg/mL)		0.45	0.37	0.13	1.12	1.15
LOQ (μg/mL)		1.50	1.23	0.43	3.73	3.83
Accuracy	I. *	99.4%	99.1%	99.8%	99.1%	99.6%
	II. (0.15%)	99.9%	100.3%	100.2%	100.3%	98.8%
	III. (0.3%)	99.6%	98.2%	99.6%	98.2%	100.1%
Intraday precision (RSD%)	I. *	0.8%	0.9%	0.3%	0.6%	1.4%
	II. (0.15%)	0.8%	1.2%	0.1%	0.9%	1.3%
	III. (0.3%)	0.5%	0.1%	0.1%	1.1%	2.0%
Intermediate precision (RSD%)	I. *	0.9%	1.5%	0.3%	1.5%	1.1%
	II. (0.15%)	1.3%	0.2%	0.2%	1.8%	2.0%
	III. (0.3%)	0.6%	0.2%	0.2%	1.5%	2.1%

* Level I.: 0.05% for IMP-A, R-ket and IMP-C; 0.1% for Ester-a and Ester-b.

**Table 4 ijms-25-02697-t004:** Content of impurities in investigated dexketoprofen formulations (*n* = 3).

Product	R-Ketoprofen	Impurity A	Sum of All Impurities
Dekenor	0.32% ± 0.04	<0.05%	0.45% ± 0.05
Ketodex	0.15% ± 0.01	<0.05%	0.57% ± 0.07

**Table 5 ijms-25-02697-t005:** Calculated thermodynamic parameters on Lux amylose-2 column.

Temperature Range (°C)	Equation	r^2^	Δ(ΔH°) (kJ/mol)	Δ(ΔS°) (J/molK)	Δ(ΔG°) (kJ/mol)	T_iso_ (°C)
10–40	lnα = 1573.7x − 5.017	0.9926	−13.1	−41.7	−0.7	40.7
40–55	lnα = −1239.5x + 3.957	0.9997	16.3	32.9	0.5	40.2

## Data Availability

The original contributions presented in this study is included in the article/Appendix A; further inquiries can be directed to the corresponding author.

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
