# Peer review of "Simultaneous Determination of Enantiomeric Purity and Organic Impurities of Dexketoprofen Using Reversed-Phase Liquid Chromatography—Enhancing Enantioselectivity through Hysteretic Behavior and Temperature-Dependent Enantiomer Elution Order Reversal on Polysaccharide Chiral Stationary Phases"

_ijms, 2024, doi:10.3390/ijms25052697_

Round 1

Reviewer 1 Report

Comments and Suggestions for Authors

Separation of enantiomers seems to be important application of contemporary chromatographic methods. Formally the manuscript is devoted to the very practical problem – the separation of ketoprofen enantiomers. However, at the same time it is full of interesting observations important for solving the related problems. They are, at first, unusual shape of the dependence of retention factor vs. acetonitrile-methanol proportion in their mixture (Fig. 3C). Secondly, it is the strong temperature dependence of enantiomers’ elution order. And, finally, it is detection of hysteresis phenomena in PO and RP modes. The combination of all these factors explains the feasibility of publication of this manuscript in IJMS.

However, before that some moments should be corrected.

At first, it is the title of the manuscript, namely such fragment as “… enantiomeric and organic impurities …”. Such dissimilar adjectives cannot be joined by the conjunction “and”. It looks like the name of a shop “men’s, women’s and rubber shoes”. Hence, the manuscript title should be slightly corrected.

All structural formulas of ketoprofen and its impurities at page 4 are drawn with fragment “CH” instead of “CH3”. Obviously, it should be corrected, as well.

Table 3, page 10: the values of “Accuracy”, “Intraday precision” and “Intermediate precision” should not be given with two decimal digits. All values should be rounded up to only one of them.

Table 5, page 14: the values Tiso, (H) and (S) should be rounded up to one decimal digit, as well.

Besides that, two examples of inappropriate word hyphenation are marked in the text of manuscript (the file with corrections is attached).

Comments on the Quality of English Language

No comments

Author Response

We greatly appreciate the remarks of Reviewer 1. We agree with the suggestion of the Reviewers, so we prepared our revised version of the manuscript according to the recommendations.

In the response remarks, questions, etc. are in italics; our responses, corrections, etc. are in normal format.

 Separation of enantiomers seems to be important application of contemporary chromatographic methods. Formally the manuscript is devoted to the very practical problem – the separation of ketoprofen enantiomers. However, at the same time it is full of interesting observations important for solving the related problems. They are, at first, unusual shape of the dependence of retention factor vs. acetonitrile-methanol proportion in their mixture (Fig. 3C). Secondly, it is the strong temperature dependence of enantiomers’ elution order. And, finally, it is detection of hysteresis phenomena in PO and RP modes. The combination of all these factors explains the feasibility of publication of this manuscript in IJMS.

However, before that some moments should be corrected.

At first, it is the title of the manuscript, namely such fragment as “… enantiomeric and organic impurities …”. Such dissimilar adjectives cannot be joined by the conjunction “and”. It looks like the name of a shop “men’s, women’s and rubber shoes”. Hence, the manuscript title should be slightly corrected.

Based on your remarks the new title is: “Simultaneous determination of enantiomeric purity and organic impurities of dexketoprofen using reversed-phase liquid chromatography. Enhancing enantioselectivity through hysteretic behavior and temperature-dependent enantiomer elution order reversal on polysaccharide chiral stationary phases”

All structural formulas of ketoprofen and its impurities at page 4 are drawn with fragment “CH” instead of “CH3”. Obviously, it should be corrected, as well.

Thank you for your remarks. Figure 1 was corrected.

Table 3, page 10: the values of “Accuracy”, “Intraday precision” and “Intermediate precision” should not be given with two decimal digits. All values should be rounded up to only one of them.

Corrected.

Table 5, page 14: the values TisoD(DH°) and D(DS°) should be rounded up to one decimal digit, as well.

 Corrected.

Besides that, two examples of inappropriate word hyphenation are marked in the text of manuscript (the file with corrections is attached).

Both word hyphenations were corrected.

Reviewer 2 Report

Comments and Suggestions for Authors

Presented study is dedicated to the simultaneous determination of dexketoprofen and several impurities using reversed-phase liquid chromatography on chiral stationary phases. The authors not only successfully achieved separation, but also demonstrated a hysteresis effect that is rare in the reversed-phase mode.

 I have several comments requiring authors’ attention.

1) Table 2 – I don’t quite understand, why are all experiments duplicated? For example: lines 35-36, 37-38, 39-40. Also, I don’t understand the logic of the table's structure, factors 1 and 2 are listed randomly, so it’s hard to estimate its influence on resolution and retention.

2) Table 3 heading – should it be “R-ket” instead of “S-ket”? As I understand, impurity is R-ketoprofen.

3) In the table 3 the authors have stated that linear range is 0.05-0.3% regarding a target concentration of 5 mg/mL of dexketoprofen. If my calculations are correct, it’s equal to 2.5-15 mg/L. However, LOQs for ketoprofen esters are about 3.7 mg/L, therefore, the linear range should be narrower.

4) Figure 3. Please, specify the column used for hysteresis investigation. Also, what is k1 here, retention factor of the first-eluting enantiomer?

5) Section 3.1 – is it necessary to mention the formic acid? All experiments were conducted using acetic acid.

6) Section 3.2. Please, specify analytical wavelength. Also, were all four columns used for hysteresis effect investigation?

Also I’ve found several typos:

Line 36: “(v/v/v). 1 mL/min” – I believe, here should be comma instead of dot.

Line 253: “3 – S-ketoprofen 4- impurity A” – missed comma.

Line 349: “CAS: 60658-04-0), , ACN” – extra comma. 

Author Response

We greatly appreciate the remarks of Reviewer 2. We agree with the suggestion of the Reviewers, so we prepared our revised version of the manuscript according to the recommendations.

In the response remarks, questions, etc. are in italics; our responses, corrections, etc. are in normal format.

Presented study is dedicated to the simultaneous determination of dexketoprofen and several impurities using reversed-phase liquid chromatography on chiral stationary phases. The authors not only successfully achieved separation, but also demonstrated a hysteresis effect that is rare in the reversed-phase mode.

 I have several comments requiring authors’ attention.

1) Table 2 – I don’t quite understand, why are all experiments duplicated? For example: lines 35-36, 37-38, 39-40. Also, I don’t understand the logic of the table's structure, factors 1 and 2 are listed randomly, so it’s hard to estimate its influence on resolution and retention.

We utilized a full factorial design methodology with Stat-Ease Design-Expert v7.0.0 software to determine optimal conditions. Two parallel sets were analyzed for enhanced evaluation, the software randomly shuffles points. The table presents the original input data, crucial for method optimization, while the influence of temperature and ACN% on resolution and retention is depicted in Figure S2. For better understanding the revised version of the manuscript supplemented as follows:

“The experimental matrix and the obtained results are summarized in Table 2. The table presents the original input data with two parallel sets, which were shuffled randomly to ensure robustness.”

2) Table 3 heading – should it be “R-ket” instead of “S-ket”? As I understand, impurity is R-ketoprofen.

Thank you for your remarks. It is corrected throughout the manuscript.

3) In the table 3 the authors have stated that linear range is 0.05-0.3% regarding a target concentration of 5 mg/mL of dexketoprofen. If my calculations are correct, it’s equal to 2.5-15 mg/L. However, LOQs for ketoprofen esters are about 3.7 mg/L, therefore, the linear range should be narrower.

Thank you for your remark. Table 3 corrected according to the reviewer`s comment.

4) Figure 3. Please, specify the column used for hysteresis investigation. Also, what is k1 here, retention factor of the first-eluting enantiomer?

For hysteresis investigation all the four columns (Lux Amylose-1, Lux Amylose-2, Lux Cellulose-1, Lux Cellulose-2) were investigated.

Yes, k1 is the retention factor of the first-eluting enantiomer.

The revised version of the manuscript supplemented based on this remark.

5) Section 3.1 – is it necessary to mention the formic acid? All experiments were conducted using acetic acid.

As mentioned in line 175, formic acid was also tested. However, acetic acid yielded better results than formic acid.

6) Section 3.2. Please, specify analytical wavelength. Also, were all four columns used for hysteresis effect investigation?

230 nm was used throughout the study. The revised version of the manuscript was supplemented with the applied detection wavelength.

Also I’ve found several typos:

Line 36: “(v/v/v). 1 mL/min” – I believe, here should be comma instead of dot.

Corrected.

Line 253: “3 – S-ketoprofen 4- impurity A” – missed comma.

Corrected.

Line 349: “CAS: 60658-04-0), , ACN” – extra comma. 

Corrected.

Reviewer 3 Report

Comments and Suggestions for Authors

In the current manuscript Tóth et al. described attempts at determination of enantiomeric and organic contamination of dexketoprofen by the use of reversed-phase HPLC technique. Thus, the main question addressed by the research constitutes the optimization of HPLC conditions in order to achieve the desired enantioseparation. It has been shown that Lux Amylose-2 was the most effective column with 0.1% (v/v) acetic acid in a ACN/water 50/50 (v/v) mixture. As a person directly involved in research related to determining the optical purity of organic compounds (especially enantiomeric excess) using the high-performance liquid chromatography technique using chiral-filled columns, I appreciate the research described in the manuscript and, above all, the effort involved in the painstaking selection of favorable separation conditions. The described research on determining enantiomeric purity and the content of other organic impurities in the tested compounds is not new, but it fits perfectly into current research trends in modern organic synthesis using tools such as high-performance liquid chromatography on chiral columns.

The manuscript was prepared with great attention to its editorial aspect and graphic design. Perhaps it would be worth going through it again to eliminate any editorial mistakes, e.g. 'deinoized water' - page 15, line 350.  The reviewer does not suggest any further improvements regarding the description of the research methodology or the description of the experiments performed. The discussion of the results included in the manuscript is extensive and clear, including conclusions being consistent with the evidence. The cited references were carefully selected and are appropriate. I hereby recommend the manuscript for publication in International Journal of Molecular Sciences in its current form.

Author Response

We appreciate the reviewer's thorough evaluation of our manuscript, particularly their recognition of our efforts in optimizing HPLC conditions for enantioseparation of dexketoprofen. Their feedback on the alignment with current research trends and the meticulous attention to editorial aspects is invaluable. We will ensure to address the suggested proofreading points before resubmission.